# GEN-Z: GENERATIVE ZERO-SHOT TEXT CLASSIFICATION WITH CONTEXTUALIZED LABEL DESCRIPTIONS

**Sachin Kumar***
Allen Institute for AI
Seattle, WA
sachink@allenai.org

**Chan Young Park**
Carnegie Mellon University
Pittsburgh, PA
chanyoun@cs.cmu.edu

**Yulia Tsvetkov**
University of Washington
Seattle, WA
yuliats@cs.washington.edu

## ABSTRACT

Language model (LM) prompting—a popular paradigm for solving NLP tasks—has been shown to be susceptible to miscalibration and brittleness to slight prompt variations, caused by its discriminative prompting approach, i.e., predicting the label given the input. To address these issues, we propose GEN-Z—a **gen**erative prompting framework for **z**ero-shot text classification. GEN-Z is generative, as it measures the LM likelihood of input text, conditioned on natural language descriptions of labels. The framework is multivariate, as label descriptions allow us to seamlessly integrate additional contextual information about the labels to improve task performance. On various standard classification benchmarks, with six open-source LM families, we show that zero-shot classification with simple contextualization of the data source of the evaluation set consistently outperforms both zero-shot and few-shot baselines while improving robustness to prompt variations. Further, our approach enables personalizing classification in a zero-shot manner by incorporating author, subject, or reader information in the label descriptions.[1]

## 1 INTRODUCTION

Language models, trained only on raw text, have been shown to perform new tasks simply by conditioning on a handful of demonstrations (Brown et al., 2020). However, *how* language models acquire this ability, known as in-context learning (ICL), is a subject of debate (Xie et al., 2022; Ahuja et al., 2023; Hahn & Goyal, 2023; Zhang et al., 2023; von Oswald et al., 2023; Wang et al., 2023) with several studies suggesting that it merely serves as a way to prime the model with the domain, concepts, or topics and the format of the target task (Min et al., 2022b; Wang et al., 2023). Furthermore, ICL has been shown to be very sensitive to the choice of training examples, their order and format in the prompt (Lu et al., 2022; Sorensen et al., 2022; Sclar et al., 2024) requiring major human effort to achieve optimal performance. In this work, we ask, "If the right demonstrations are challenging to find and only serve to *implicitly* prime the model, can we achieve the same performance zero-shot if we prime the language model *explicitly*?"

We introduce GEN-Z, a robust zero-shot generative prompting framework for text classification (Figure 1) which achieves results on par with in-context learning with much better stability in performance. Our approach consists of two key ideas. First, most text classification methods follow a discriminative setup, which involves estimating the probability of the labels given the input, which can be sensitive to prompt or verbalizer variations. Instead, we use a generative setup, which involves estimating the probability of generating the input given different labels, which has been shown to have better worst-case performance (Min et al., 2022a). Second, to prime the models to solve the task, we propose to explicitly incorporate contextual information via expressive label descriptions. We first generate a description for each label that captures various factors that can influence the label and then estimate the probability of generating the input text given the label description (e.g. "This Reddit post contains hate speech about race" for hate speech detection where the data source "Reddit" and the subject "race" are additional factors). Finally, to further reduce variance from different label

---

*Part of this work was done when Sachin was a PhD student at Carnegie Mellon University.
[1]We provide the code to reproduce our results at: https://github.com/Sachin19/generative-classification/

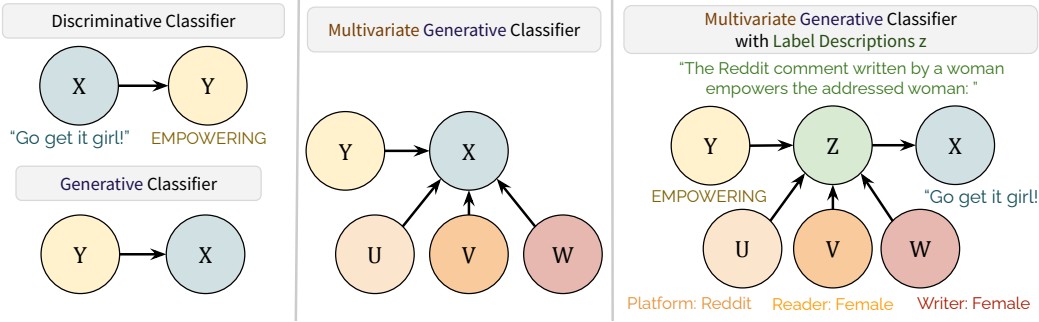

Figure 1: Illustration of the proposed zero-shot generative text classifier with label description and examples.

description phrasings, we propose to compute and aggregate the results across multiple paraphrases of the label descriptions.

We evaluate GEN-Z by conducting experiments with six open-source language model families (GPT2, OPT, Pythia, GPT-J, Llama, and Llama2) with models ranging from 125M to 13B parameters on 19 semantic text classification tasks (comprising sentiment, topic, hate speech, and emotion classification). We show that incorporating readily available additional variables like text source, domain, subject, information of author, audience, and the addressee of the text, in our approach leads to substantial improvements compared to vanilla zero-shot baselines, strong retrieval based zero-shot methods, and performs on par with heavily tuned in-context learning methods.

## 2 ZERO-SHOT GENERATIVE CLASSIFICATION WITH LABEL DESCRIPTIONS

This section describes our proposed method. First, we motivate our zero-shot setup by drawing connections to interpretations of in-context learning as concept learning. We then give an overview of generative classification followed by an explanation of how we incorporate contextual information into the model using label descriptions.

### 2.1 IN-CONTEXT LEARNING AS IMPLICIT CONCEPT LEARNING

We seek to build a probabilistic classifier, $p(y|\mathbf{x})$ that takes text $\mathbf{x}$ as input and predicts $y \in \mathcal{Y}$, the set of all labels. Language models trained to predict the next token given the history have been shown to be able to perform classification tasks in-context without fine tuning (Brown et al., 2020). Given $k$ demonstrations $\{(\mathbf{x}_1, y_1), \ldots, (\mathbf{x}_k, y_k)\}$ and a test example $\mathbf{x}$, the label can be predicted using, $p_{\text{LM}}(y|\mathbf{x}_1, y_1, \ldots, \mathbf{x}_k, y_k, \mathbf{x})$. In practice, the label is verbalized in natural language (e.g., the words "negative" and "positive" for sentiment classification). Prior work (Xie et al., 2022) has shown evidence that in-context learning implicitly performs Bayesian inference where this probability can written as the following marginalization,

$$p_{LM}(y|\mathbf{x}_1, y_1, \ldots, \mathbf{x}_k, y_k, \mathbf{x}) = \int_{\Theta} p(y|\mathbf{x}_1, y_1, \ldots, \mathbf{x}_k, y_k, \mathbf{x}, \boldsymbol{\theta}) p(\boldsymbol{\theta}|\mathbf{x}_1, y_1, \ldots, \mathbf{x}_k, y_k, \mathbf{x}) d\boldsymbol{\theta}$$

$$= \int_{\Theta} p(y|\mathbf{x}, \boldsymbol{\theta}) p(\boldsymbol{\theta}|\mathbf{x}_1, y_1, \ldots, \mathbf{x}_k, y_k, \mathbf{x}) d\boldsymbol{\theta}$$

where $\boldsymbol{\theta} \in \Theta$ represents a concept or a topic variable, on in general, context required to solve the task. Following Wang et al. (2023), we also make a simplifying assumption that the test example $\mathbf{x}$ is independent of the sampling of the demonstrations, so $y$ is independent of the demonstrations given $\theta$ and $\mathbf{x}$. That is, the context variable $\boldsymbol{\theta}$ acts like an approximate sufficient statistic for the posterior information related to the demonstrations. The variables are latent and the model is expected to implicitly figure out the context needed to solve the task from the given demonstrations. Intuitively, $p(\boldsymbol{\theta}|\ldots)$ concentrates on the concept mentioned in the demonstrations, that is, the LM softly predicts this concept. In this work, we take this formulation to the extreme by defining it as a Dirac delta distribution concentrated on the right concept, that is given the right concept $\theta^*$, we set $p(\theta^*|\ldots) = 1$

and zero everywhere else. For semantic text classification tasks, we argue that the right concepts can be specified in natural language and hence, the demonstrations may not be necessary. The label prediction probability is thus reduced to,

$$p_{\text{LM}}(y|\mathbf{x}_1, y_1, \ldots, \mathbf{x}_k, y_k, \mathbf{x}) \approx p_{\text{LM}}(y|\mathbf{x}, \theta^*)$$

This describes zero-shot inference **contextualized** on the concepts. In this work, we experiment with many different kinds of contexts and their influence on text classification performance such as those describing the domain, source, author, or audience of the input text among others (§3).

## 2.2 MULTIVARIATE GENERATIVE CLASSIFICATION

The approach discussed so far describes a discriminative classifier, which predicts the label as $\hat{y} = \arg\max_{y_i \in \mathcal{Y}} p(y_i|\mathbf{x})$. They are designed to distinguish the correct label among possible choices. A generative classification framework reinterprets this objective using Bayes' rule and a different factorization as

$$\hat{y} = \arg\max_{y_i} \frac{p(\mathbf{x}, y_i)}{p(\mathbf{x})}$$
$$= \arg\max_{y_i} p(\mathbf{x}|y_i)p(y_i)$$

Here the denominator $p(\mathbf{x})$ is independent of the label and can be ignored. Further, assuming equal prior probability of all labels, $p(y_i)$ can also be ignored making the classification objective $\arg\max_{y_i} p(\mathbf{x}|y_i)$. In a generative setup, we assume a label is generated first (e.g., an author decides to write a negative review), and then the text (e.g., the negative review) is produced conditioned on the label. Prior work has shown evidence that generative classifiers can be more robust than the discriminative ones which may look for shortcuts to predict the label (Yogatama et al., 2017).

To incorporate contextual information $\theta^*$, we propose multivariate generative classification which generalizes it to more variables that might influence the generative process of the input text, expressing the generative probability of $\mathbf{x}$ as $p(\mathbf{x}|y, u, v, \ldots)$, where $\theta^* = u, v, \ldots$ represent the additional factors. For example, to generate a review, not only is the author influenced by the polarity but also by the item they review, the medium where they write the review, their target audience, their writing style, and so on. Similar context can also be added to a discriminative classifier $p(y|\mathbf{x}, u, v, \ldots)$ which is one of our baselines.

## 2.3 CONTEXTUALIZED LABEL DESCRIPTIONS

In practice, as introduced in (Brown et al., 2020), language models can be used in a zero-shot setup by computing $p_{\text{LM}}(z(y_i)|\mathbf{x})$ or in our case, $p_{\text{LM}}(\mathbf{x}|z(y_i, u, v, \ldots))$. Here, $z(\cdot)$ is referred to as a verbalizer which expresses the label in natural language form so that meaningful probabilities can be computed. In this work, since the verbalizers are only concerned with the label, we refer to them as *label descriptions*. A simple example is "This is terrible." and "This is amazing." for negative and positive label respectively. The choice of this description, however, can lead to large variance in the model performance with downstream classification performance can range from near perfect to near chance (Liu et al., 2023; Holtzman et al., 2021; Zhao et al., 2021).

To reduce this variance, we propose to use multiple variations of the descriptions $z$. More formally, we modify the generative story as: the labels and other contextual variables generate label descriptions $z$ which then inform the generation of the text (see Figure 1),

$$p(y_i|\mathbf{x}, u, v, \ldots) \propto p(\mathbf{x}, y_i|u, v, \ldots)$$
$$= \sum_{z_i \in \mathcal{Z}(y_i, u, v, \ldots)} p(\mathbf{x}, y_i, z_i|u, v, \ldots)$$
$$= \sum_{z_i \in \mathcal{Z}(y_i, u, v, \ldots)} p(x|y_i, z_i, u, v, \ldots)p(z|y_i, u, v, \ldots)p(y_i|u, v, \ldots)$$

Here, $\mathcal{Z}(y_i, u, v, \ldots)$ denotes the set of all ways to describe the label $y_i$ and the context in natural language[2]. $p(z|y_i, u, v, \ldots)$ measures the existence probability of the description[3]. Since each

---

[2]We show that $\sim$10 diverse paraphrases of the description are sufficient to obtain good performance.

[3]Note that we are not measuring grammatical plausibility of a description, hence measure $p(z|\cdot)$ using an LM is not appropriate in this setting.

description is equally plausible to exist given the label $y_i$ and other variables, we drop $p(z_i|y_i, u, v, \dots)$ (see Figure 1 right). Further, assuming independence of the label and the contextual factors[4] and equal prior likelihood of all labels, we also drop $p(y_i|u, v, \dots)$.[5] Hence, the first term in the summation can be reduced to $\sum_{z_i} p(\mathbf{x}|z_i)$, which is our inference objective, where we evaluate the probabilities using the conditional probabilities of the LM. We compute this term for each label under consideration $y_i$ and predict the label which obtains the highest value. Notably, unlike common prompting scenarios, the label descriptions, $z_i$, are unique for each label $y_i$ being considered and can be specialized by adding any available information about the instance in natural language format. We refer to our approach as GEN-Z for **Gen**erative **Z**ero-Shot Classification.

## 3 EXPERIMENTAL SETUP

**Datasets, Models, and Label Descriptions** We evaluate on 18 text classification datasets encompassing diverse tasks, domains, and difficulty levels. These datasets include varying numbers of classes and attributes that can be used as additional context to improve the classification performance. We consider all contexts that were provided with each dataset. We consider the following tasks divided in to two groups: (1) sentiment, topic, and hate speech detection which in addition to the input text are accompanied by information about the domain, source or subject of the input text. For example, hate speech datasets which contain information about the source (such as Reddit or Twitter) and the subject of hate (such as national origin or race); and (2) politeness, and empowerment prediction which are pragmatic tasks that depend on social variables such as demographic information of the author, addressee, or the reader (such as gender, age, educational background, etc.). Table 4 in the appendix summarizes the details of each dataset we use. We measure performance using publicly available validation or test sets, without using the training data at all. We experiment with the six classes of open-source models: GPT2 (Small, Medium, Large, and XL) (Radford et al., 2019), OPT (Zhang et al., 2022)(1.4B and 2.7B), Pythia (Biderman et al., 2023) (1.4B, 2.8B and 6.7B), GPT-J (6B) (Wang & Komatsuzaki, 2021), Llama 1 (Touvron et al., 2023a) (7B and 13B) and Llama 2 (Touvron et al., 2023b) (7B and 13B).[6] All these models are pretrained on only raw text without additional fine-tuning on supervised datasets.[7]

For each task, we manually write one minimal label description per label using a template (see complete list in Table 5). We then generate 20 paraphrases of each label description by querying ChatGPT.[8] This process needs to be done only once for each task and, in practice, any paraphrasing model can be employed. We further manually verify the correctness of each paraphrase. For each dataset, we run the evaluation ten times where in each run we subsample $1 \leq n \leq 10$ paraphrases from this set. We evaluate all methods using macro-F1 score and report mean and standard deviation across these runs.

**Baselines** We compare GEN-Z with the following zero-shot baselines.

- **Discriminative** methods predict the label using $\sum_{z_i} p(z_i|\mathbf{x})$. We consider three versions of this baseline. DISC-SINGLE-NC predicts the label with **n**o **c**ontext (the context information is removed from $z_i$) and only one description is considered. This is the simplest zero-shot setup that most prior work considers canonical. DISC-SINGLE predicts the label using $p(z_i|\mathbf{x})$ where $z_i$ corresponds to only one description. DISC-MULTIPLE predicts the label using $\sum_{z_i} p(z_i|\mathbf{x})$ which is the discriminative version of our proposed method. For the last two baselines, we further have three

---

[4]The true prior probability of any label is unlikely to depend on the contextual factors like the domain of the text, or the personal attributes of the user reading, writing, or being described in the text.

[5]While we make this assumption for simplification, future work may consider non-uniform priors to further improve performance.

[6]We do not report results with 70B sized models to due to its high computational requirements for the scale of our experiments. While quantization (Dettmers et al., 2023) approaches have been proposed to run models of this scale on consumer hardware, in our initial exploration such approaches vastly underperformed 16-bit versions for our experiments. Further, our budget prohibits us from experimenting with closed-source models like GPT3 which according to Lyu et al. (2023) can cost more than 4500 USD for the scale of our experiments. We leave these explorations for future work.

[7]Instruction-tuned models have shown to also perform well in-context but they are trained primarily as discriminative classifiers and thus cannot be used for generative classification making comparisons unfair.

[8]We used the free tier of ChatGPT for this purpose: https://chat.openai.com/chat.

Table 1: Zero-shot Macro-F1 with GPT-J (6B). We report average$_{\text{std}}$ over 10 runs. Our proposed approach is GEN-Z. More results are provided in Appendix C. The baseline names are shortened with their initials.

| | DS-N | DS | DM | DP-N | DP | DPM | GS-N | GS | GEN-Z |
|---|---|---|---|---|---|---|---|---|---|
| SST2 | $69.5_{(1.2)}$ | $65.9_{(0.9)}$ | $43.6_{(1.0)}$ | $74.5_{(1.2)}$ | $75.7_{(1.3)}$ | $72.3_{(0.7)}$ | $80.0_{(1.0)}$ | $87.1_{(0.6)}$ | $\mathbf{91.7}_{(0.2)}$ |
| SST5 | $18.6_{(0.7)}$ | $24.2_{(0.9)}$ | $25.1_{(0.2)}$ | $26.2_{(0.7)}$ | $29.8_{(0.6)}$ | $35.6_{(0.2)}$ | $34.8_{(0.8)}$ | $36.0_{(1.2)}$ | $\mathbf{40.9}_{(0.5)}$ |
| Yelp2 | $68.1_{(1.0)}$ | $64.2_{(1.1)}$ | $76.7_{(0.3)}$ | $70.5_{(0.5)}$ | $70.2_{(0.6)}$ | $75.9_{(0.5)}$ | $76.1_{(0.8)}$ | $85.4_{(0.7)}$ | $\mathbf{89.9}_{(0.1)}$ |
| Yelp5 | $24.0_{(1.1)}$ | $26.2_{(0.7)}$ | $25.6_{(0.3)}$ | $30.5_{(1.1)}$ | $28.5_{(1.4)}$ | $32.0_{(0.3)}$ | $35.5_{(0.8)}$ | $38.0_{(0.7)}$ | $\mathbf{42.0}_{(0.3)}$ |
| MR | $67.7_{(0.9)}$ | $63.3_{(1.0)}$ | $51.3_{(0.5)}$ | $72.3_{(1.0)}$ | $70.6_{(1.2)}$ | $74.6_{(0.5)}$ | $76.0_{(0.9)}$ | $84.2_{(0.6)}$ | $\mathbf{87.0}_{(0.2)}$ |
| CR | $57.3_{(2.7)}$ | $57.1_{(2.8)}$ | $51.8_{(1.1)}$ | $60.3_{(2.1)}$ | $66.4_{(2.9)}$ | $66.7_{(0.9)}$ | $72.9_{(1.7)}$ | $83.8_{(1.5)}$ | $\mathbf{87.0}_{(0.2)}$ |
| Tweet3 | $36.0_{(0.2)}$ | $35.5_{(0.4)}$ | $31.9_{(0.1)}$ | $39.4_{(0.4)}$ | $39.7_{(0.5)}$ | $\mathbf{43.0}_{(0.1)}$ | $42.2_{(0.3)}$ | $42.2_{(0.3)}$ | $41.1_{(0.1)}$ |
| FP | $36.6_{(2.3)}$ | $32.3_{(2.0)}$ | $25.2_{(0.5)}$ | $38.7_{(1.9)}$ | $39.8_{(3.2)}$ | $47.8_{(1.4)}$ | $44.7_{(1.8)}$ | $48.2_{(2.4)}$ | $\mathbf{52.9}_{(1.1)}$ |
| PS | $38.0_{(5.1)}$ | $32.9_{(4.9)}$ | $30.3_{(1.6)}$ | $35.8_{(3.4)}$ | $33.8_{(4.1)}$ | $21.3_{(1.3)}$ | $39.2_{(3.3)}$ | $38.4_{(4.2)}$ | $\mathbf{42.4}_{(1.2)}$ |
| AGNews | $34.8_{(0.5)}$ | $34.8_{(0.5)}$ | $37.9_{(0.2)}$ | $54.6_{(0.5)}$ | $54.6_{(0.5)}$ | $72.0_{(0.2)}$ | $64.9_{(0.4)}$ | $64.9_{(0.4)}$ | $\mathbf{77.0}_{(0.1)}$ |
| DBPedia | $42.5_{(0.5)}$ | $39.4_{(0.7)}$ | $32.8_{(0.4)}$ | $61.7_{(0.5)}$ | $66.6_{(0.8)}$ | $78.9_{(0.1)}$ | $71.7_{(0.5)}$ | $71.7_{(0.5)}$ | $\mathbf{80.1}_{(0.2)}$ |
| HS18 | $17.4_{(0.5)}$ | $14.7_{(0.7)}$ | $10.1_{(0.0)}$ | $37.2_{(1.0)}$ | $38.7_{(0.9)}$ | $29.9_{(0.3)}$ | $50.2_{(0.7)}$ | $55.8_{(0.9)}$ | $\mathbf{62.6}_{(0.3)}$ |
| Ethos (NO) | $50.8_{(3.0)}$ | $50.8_{(3.0)}$ | $\mathbf{63.2}_{(3.0)}$ | $54.9_{(2.1)}$ | $54.9_{(2.1)}$ | $55.7_{(1.5)}$ | $50.4_{(3.0)}$ | $50.4_{(3.0)}$ | $56.3_{(0.8)}$ |
| Ethos (SO) | $33.8_{(4.7)}$ | $33.8_{(4.7)}$ | $20.4_{(0.7)}$ | $52.5_{(5.4)}$ | $52.5_{(5.4)}$ | $55.3_{(1.7)}$ | $52.7_{(2.8)}$ | $52.7_{(2.8)}$ | $\mathbf{62.3}_{(1.6)}$ |
| Ethos (Race) | $27.4_{(3.8)}$ | $27.4_{(3.8)}$ | $15.5_{(0.0)}$ | $54.5_{(4.3)}$ | $54.5_{(4.3)}$ | $50.9_{(1.9)}$ | $56.1_{(3.1)}$ | $56.1_{(3.1)}$ | $\mathbf{60.5}_{(1.3)}$ |
| Ethos (Religion) | $48.2_{(3.9)}$ | $48.2_{(3.9)}$ | $42.5_{(1.8)}$ | $62.3_{(2.8)}$ | $62.3_{(2.8)}$ | $62.8_{(1.3)}$ | $62.9_{(3.7)}$ | $62.9_{(3.7)}$ | $\mathbf{70.1}_{(0.9)}$ |
| Emotions | $24.9_{(0.4)}$ | $23.5_{(0.6)}$ | $29.0_{(0.3)}$ | $31.4_{(1.0)}$ | $31.8_{(1.1)}$ | $\mathbf{37.8}_{(0.2)}$ | $30.3_{(0.8)}$ | $30.3_{(0.8)}$ | $32.7_{(0.3)}$ |

versions of each which differ in how the context is provided (only in the label, before the input, and before the input as an instruction; more details in Appendix B. We report results with the first version in the main paper as it performs the best of the three.).

- **Calibrated discriminative** methods use $p(z_i|x)/p(z_i|\text{NULL})$ for label inference (we use BOS token for the corresponding LMs as NULL). Since language model probabilities can be poorly calibrated and suffer from competition between different label descriptions with the same meaning, this method relies on pointwise mutual information (PMI) between $x$ and $y$ to make a prediction (Holtzman et al., 2021). We again consider three versions of this setup: without context (DISC-PMI-NC), with context but only one label description (DISC-PMI), and with context and multiple label descriptions (DISC-PMI-MULTIPLE). The last one is the calibrated discriminative version of our proposed method.
- **Generative** baselines predicts the label using $p(\mathbf{x}|z_i)$, that is using only one label description. We consider two versions of this baseline, one without context (GEN-SINGLE-NC) and one with context (GEN-SINGLE) in the label description. Both of these are an ablation of our proposed method. The former method (without context) also describes the method proposed in Min et al. (2022a).

In addition, we show comparisons with the following baselines which incorporate context implicitly either via few-shot examples or using retrieval based techniques on unlabeled data. For these baselines, we experiment with 8- and 16-shot settings and report the best of the two.

- **ICL** describes in-context learning baselines. We consider four versions of this baseline: (a) ICL-DISC, a simple discriminative method which compares probabilities of label descriptions, (b) ICL-GEN, a generative baseline with the exact same setup (Min et al., 2022a), (c) ICL-PMI which calibrates the probabilities same as DISC-PMI, (d) ICL-DC which is another discriminative calibration method introduced in Fei et al. (2023b).
- **Z-ICL** (Lyu et al., 2023) describes a psuedo-demonstration based setup where unlabeled texts are sampled from a corpus and assigned random labels. This setup is zero-shot but still requires access to a corpus. For this setup, we reproduce the setup proposed in  and report both discriminative and generative results.

## 4 RESULTS

We categorize the results into two groups: domain-aware classification, which considers the domain of the text as an additional factor, and personalized classification, which includes personal attributes of writers and readers as additional factors.

Table 2: Best of 8/16 shot baselines vs Gen-Z (zero-shot). We report average$_\text{std}$ over 5 seeds.

| | ICL (Disc) | ICL (CC) | ICL (DC) | ICL (Gen) | Z-ICL (Disc) | Z-ICL (Gen) | GEN-Z (Ours) |
|---|---|---|---|---|---|---|---|
| SST2 | $91.0_{(6.0)}$ | $90.8_{(3.2)}$ | $\mathbf{94.0}_{(1.3)}$ | $88.8_{(1.3)}$ | $82.6_{(0.2)}$ | $82.6_{(0.2)}$ | $91.7_{(0.2)}$ |
| CR | $81.4_{(6.9)}$ | $86.5_{(0.8)}$ | $87.0_{(4.0)}$ | $84.4_{(2.8)}$ | $78.8_{(0.4)}$ | $80.1_{(0.1)}$ | $\mathbf{87.0}_{(0.2)}$ |
| MR | $\mathbf{93.1}_{(0.7)}$ | $91.3_{(1.1)}$ | $93.1_{(0.5)}$ | $84.0_{(6.8)}$ | $81.0_{(0.3)}$ | $81.9_{(0.7)}$ | $87.0_{(0.2)}$ |
| SST5 | $\mathbf{42.9}_{(0.9)}$ | $40.8_{(5.4)}$ | $40.3_{(4.9)}$ | $42.9_{(0.9)}$ | $30.9_{(0.3)}$ | $38.7_{(0.5)}$ | $40.9_{(0.5)}$ |
| FP | $46.4_{(6.9)}$ | $46.7_{(4.2)}$ | $\mathbf{61.6}_{(3.3)}$ | $43.3_{(2.3)}$ | $44.9_{(3.0)}$ | $51.1_{(1.2)}$ | $52.9_{(1.1)}$ |
| PS | $26.6_{(6.1)}$ | $25.5_{(5.2)}$ | $31.4_{(3.0)}$ | $39.8_{(2.1)}$ | $39.5_{(4.0)}$ | $\mathbf{43.9}_{(2.7)}$ | $42.4_{(1.2)}$ |
| AGNews | $68.4_{(9.9)}$ | $76.8_{(7.2)}$ | $\mathbf{81.5}_{(5.1)}$ | $72.3_{(3.2)}$ | $67.2_{(1.3)}$ | $75.2_{(0.5)}$ | $77.0_{(0.1)}$ |
| DBpedia | $83.5_{(3.0)}$ | $90.6_{(1.7)}$ | $\mathbf{92.4}_{(1.2)}$ | $79.9_{(3.8)}$ | $61.3_{(0.6)}$ | $74.9_{(3.6)}$ | $80.1_{(0.2)}$ |
| HS18 | $51.5_{(4.7)}$ | $41.6_{(8.6)}$ | $57.3_{(2.5)}$ | $45.0_{(0.5)}$ | $43.4_{(0.6)}$ | $51.1_{(0.7)}$ | $\mathbf{62.6}_{(0.3)}$ |
| E (Religion) | $30.7_{(14.3)}$ | $28.0_{(13.8)}$ | $43.8_{(6.7)}$ | $67.9_{(1.8)}$ | $51.0_{(2.6)}$ | $51.0_{(2.6)}$ | $\mathbf{70.1}_{(0.9)}$ |
| E (NO) | $23.1_{(8.7)}$ | $18.2_{(2.1)}$ | $40.7_{(7.8)}$ | $37.6_{(3.5)}$ | $37.6_{(3.5)}$ | $26.2_{(1.1)}$ | $\mathbf{56.3}_{(0.8)}$ |
| E (Race) | $36.4_{(11.8)}$ | $44.7_{(17.4)}$ | $51.4_{(6.4)}$ | $49.1_{(3.0)}$ | $46.3_{(1.2)}$ | $39.7_{(3.5)}$ | $\mathbf{60.5}_{(1.3)}$ |

## 4.1 DOMAIN-AWARE CLASSIFICATION

Table 1 shows the comparison of the performance of different zero-shot methods on sentiment, topic, emotion, and hate speech classification for GPT-J. Table 2 shows comparisons of GEN-Z with the few-shot baselines. The remaining results are reported in Appendix C.

We find that GEN-Z overall outperforms all baselines approaches in the zero-shot setting. We do not see a clear trend in the simple discriminative baselines (DISC-SIMPLE-NC, DISC-SIMPLE, and DISC-MULTIPLE) where adding contexts and multiple descriptions sometimes help and sometimes hurts performance. In the calibrated discriminative baselines, the trends become clearer. The no-context version outperforms simple discriminative baselines as it accounts for surface form competition. Adding context also helps but adding multiple label descriptions does not always improve performance. We see this trend also in our full results where we range the number of label descriptions from 1 to 10. The generative baseline without context and a single description (Min et al., 2022a, GEN-SIMPLE-NC) outperforms most discriminative approaches and adding context leads to even more improvements confirming the efficacy of our method.

Furthermore, GEN-Z in a zero-shot setting is either best or second best performing method when compared to strong few-shot baselines on sentiment and hate-speech detection (Table 1). One particular dataset where our method lags behind is Dbpedia topic classification where the label set consists of 14 classes. Our qualitative analysis reveals that most errors made by our approach correspond to three classes with semantic overlap (Album, Film, Written Work) which given our simplistic label descriptions make it difficult for the model to distinguish. This requires further investigations into the specificity of the descriptions which we leave for future work. Finally, compared to all baselines GEN-Z shows the smallest variance in performance due to prompt selection by aggregating over multiple prompt paraphrases, whereas few-shot baselines exhibit large deviations.

We additionally conduct ablation studies to assess the impact of each proposed component on performance. While some of these ablations we used for baseline comparisons in Table 1, here we give them a more thorough treatment comparing performance across multiple model sizes.

**Effect of number of label descriptions.** In this ablation, we vary the number of label descriptions over which the aggregation is perform from $l=1$ to $l=10$ and observe the change in performance. For each $l$, we do this evaluation 10 times and report the averaged mean and standard deviation across all 17 tasks. We find that in the generative classification settings, in the majority of cases, increasing the number of label descriptions improves the model performance highlighting the utility of this approach. Further, we observe that the performance starts to stabilize between $k=6$ and $k=10$ which suggests that not many descriptions are required overall. In contrast, for discriminative baselines in all three versions we considered, we observe no clear trend as increasing $k$ often results in a decrease in performance.

**Effect of additional variables.** To measure the effect of provided contextual information (domains, subject, data source), we conduct ablation by modifying the label description to exclude this information across different number of label descriptions (similar to GEN-SIMPLE-NC and DISC-SIMPLE-NC). We report the full results in Appendix C. We observe a significant drop in

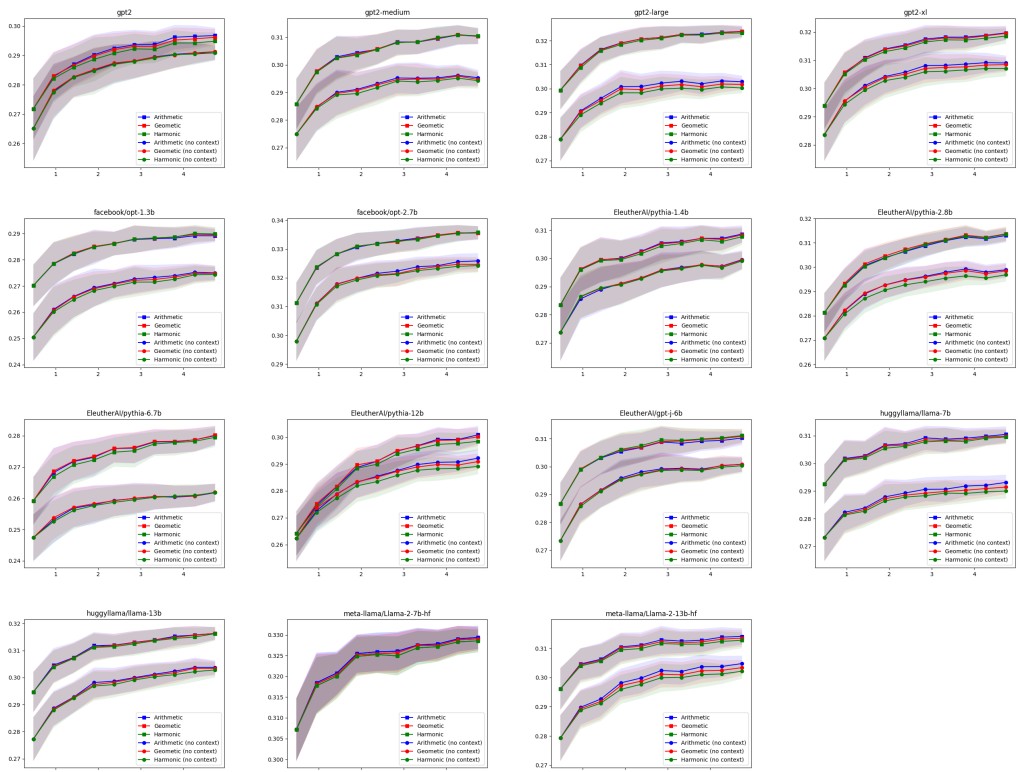

Figure 2: Full results for domain aware classification for GEN-Z. The x-axis shows number of label descriptions per label and the y-axis indicates the average accuracy across all the tasks except (Politeness and Empowerment). We conduct a thorough analysis of this setup as discussed in (section 4): removing domain information from the descriptions, different aggregation strategies as well as evaluating on different model sizes and families.

the performance across all tasks if we remove the domain or data source information including our method as well as the baselines. This drop is more significant in larger models. We hypothesize that specifying the domain information helps prime the model probabilities to the right distributional landscape allowing more meaningful comparisons between probabilities assigned to different labels. Further, we hypothesize that in a generative classification setting, the label followed by the input text, resemble natural data found in pretraining whereas it is difficult to specify this information in a discriminative setup.

**Effect of model size.** We measure if the presented results holds across model scales. We repeat the same experiment across 14 more models with size ranging from 125M to 13B. We find that with GEN-Z, across reasonably large models, going larger improves performance on average. We see substantial improvement from GPT2-M to L to XL, and Pythia 1.4B to 2.7B and 6.7B to 12B.[9] The trend is reversed for the Llama2 models where the 13B models performs slightly worse overall and warrants further investigation.

**Ablation on aggregation strategy** In GEN-Z, we aggregate the probabilities obtained using different label descriptions by simply summing them (which is the same as their arithmetic mean, for comparison purposes). This aggregation is theoretically grounded in the probabilistic framework we design (Figure 1). Prior work has considered several other aggregation strategies for ensembling model outputs that we compare with in this ablation. We compare against geometric mean (or arithmetic mean of the log probabilities, the most common way to aggregate model outputs) and harmonic mean. We find that in our proposed generative setup, the performance across three aggregation strategies is

---

[9]The performance is not strictly comparable across model families due to differences in pretraining corpora.

| Aut. \ Ad. | No | Yes | Age \ Ed. | No | Yes |
|---|---|---|---|---|---|
| No | 81.1 | 84.8 | No | 80.1 | 81.4 |
| | (1.43) | (1.83) | | (0.30) | (0.21) |
| Yes | 81.8 | **85.0** | Yes | 82.2 | **83.3** |
| | (1.07) | (2.42) | | (0.32) | (0.24) |

(a) Empowerment (F1).  (b) Politeness (Accuracy)

Table 3: Personalized classification results on GPT2-Large with our model. Each cell represents whether the demographic attribute was used in the label description or not. Aut.=Author, Ad.=Addressee, Ed.=Education.

largely similar, arithmetic mean outperforming the other two slightly overall with harmonic mean winning out in smaller models. We hypothesize that this effect is due to harmonic mean's property of ignoring outliers. Future work may analyze this strategy in-depth. For the discriminative setup, arithmetic mean almost always performs the best. Peculiarly, harmonic means shows sharp decrease in performance with increasing descriptions and warrants further investigation which we leave to future work.

## 4.2 PERSONALIZED CLASSIFICATION

In this setup, we evaluate our proposed approach on two datasets where personal information about the author, the addressee, or even the audience may affect the prediction. We experiment with two tasks: (1) empowerment prediction (Njoo et al., 2023) where given a Reddit comment, the goal is to predict whether it empowers or disempowers (or is condescending to) the addressee of the comment. We use the author's and the addressee's gender in this task[10]. (2) Politeness prediction (Pei & Jurgens, 2023) where given an email snippet the goal is to predict whether it is polite or not. We again consider binary labels. What is considered polite may vary with the reader dependent on cultural factors. This dataset consists of information about the annotator's age, gender, race, and educational background. We focus on age and educational background as they were the primary delineators of variation measured by the authors. That is given the author's age and educational background, we predict the perceived politeness of the text and sum their probabilities to make the final predictions. We do not aggregate these probabilities over each possible value of age and educational background but rather use only the ones reported in the test set. The results for both datasets for GPT2-Large are reported in Table 3[11]. We only report the results for our proposed approach with varying number of personal attributes considered, as discriminative models performed poorly in this setup (<50% accuracy across both tasks). We find that for both test sets, personalizing the predictions with demographic variables helps improve performance. For empowerment prediction, the gender of the addressee, and politeness, the age of the annotator affect the performance more than the other variables. The latter is consistent with prior studies that show cultural differences in politeness across different age groups (Pei & Jurgens, 2023).

## 5 RELATED WORK

**In-context learning** In-context learning (ICL) is the standard paradigm for prompting LMs to perform tasks (Brown et al., 2020; Liu et al., 2023).Much recent work has been done to understand why it works and how it can be improved. For example, Xie et al. (2022); Wang et al. (2023); Dai et al. (2023) have argued that it implements general-purpose learning mechanisms such as Bayesian inference or gradient descent. Min et al. (2022b) showed that for classification tasks, the input-label pairing format plays the most crucial role in ICL. We build on these findings and develop a zero-shot inference approach. While in-context learning with more example has usually performed than zero-shot inference, it comes at the cost of more token consumption and may hit the context length limit when the input and output text are long. The choice of demonstrations can lead to high variance in the model performance (Zhao et al., 2021; Fei et al., 2023a; Han et al., 2023) and prior work has investigated various demonstration selection- and ordering strategies to boost performance (Lu, 2022).

---

[10]We use binary gender here; the evaluation set does not contain any other information

[11]The results for other models can be found in Appendix C

In this work, we show that the zero-shot setting is underexplored and can surpass in-context learning for text classification tasks.

Recent work has also studied instruction following in models, either directly on a pretrained language model or by fine-tuning it to follow instructions using a collection of NLP tasks framed as instruction following tasks (Wei et al., 2022). Instructions and few-shot learning can also be used together. Again, depending on how instructions are phrased, however, can significantly alter the model outputs, even in instruction fine-tuned models (Sun et al., 2023). In contrast, several studies have also developed prompt engineering techniques, that is creating a sequence of prefix tokens or prompts that increase the probability of getting desired output given input (Liu et al., 2023). These techniques rely on available training data for each task. In this work, we focus on a zero-shot prompting setup operating in a setting where no training data for customizing classification models is available.

**Discriminative versus Generative Classification**    Text classification studies with prompting have primarily focused on discriminative classification, which focuses on constructing input prompts that get prepended to each input text to predict the classification label. That is conditioning on the input to generate the output. Generative or noisy channel models (Brown et al., 1993) have been previously investigated for various NLP tasks, such as machine translation (Yamada & Knight, 2001; Yee et al., 2019) and question answering (Lewis & Fan, 2019). Prior work has empirically demonstrated that generative models are more robust to distribution shift in text classification than discriminative models (Yogatama et al., 2017). Recently, Min et al. (2022a) explored the use of a generative model with prompting, leveraging pretrained language models for various text classification tasks. In this work, we build a multivariate generative classification by incorporating label descriptions. These descriptions capture various contextual information associated with each example, allowing for effective priming and customization of the classifier.

**Social and personal factors in NLP**    Machine learning systems have been shown to reflect and amplify social prejudices in human-written text, resulting in systemic biases in performance towards specific demographic groups (Mehrabi et al., 2021). Such classifiers learn spurious correlations between the label and the demographic information reflected in text either explicitly through their mentions in the text (such as names, sexuality, and race among others) or their writing style. These issues are exacerbated through annotation artifacts (Sap et al., 2019; 2022) or unbalanced datasets (Kiritchenko & Mohammad, 2018). Various solutions proposed in the literature aim to learn models that are fair to all demographics using methods like adversarial learning (Han et al., 2021a;b) and distributionally robust optimization (Zhou et al., 2021). A distinct but closely related motivation towards developing such solutions is user privacy—models should never use any personally identifiable attributes to make any predictions as it could lead to unintended negative consequences (Elazar & Goldberg, 2018). Ravfogel et al. (2020; 2022) propose methods to scrub demographic information from model representations given a trained model with little loss in model accuracy.

In contrast, few studies have shown that incorporating factors such as gender, age, region, or country of the authors as features can improve text classification performance (Volkova et al., 2013; Hovy, 2015; Yang & Eisenstein, 2017; Lynn et al., 2017; Huang & Paul, 2019; Ostapenko et al., 2022). Most of these studies are based on the assumption that social and personal factors are causally related to both the writing style and the target label. As a result, they treat classification as a domain adaptation problem in which demographic attributes divide the data distribution into different domains.

## 6    CONCLUSIONS

We introduce GEN-Z, a robust generative zero-shot text classification framework that seamlessly incorporates contextual information beyond the input text itself. GEN-Z leverages LM likelihood of generating the input text based on various label descriptions that reflect context, enabling more robust predictions than discriminative approaches. We evaluate our framework across two task categories: domain-aware classification and personalized classification, covering 19 diverse text classification datasets with varying tasks, domains, and difficulty levels, alongside multiple label description paraphrases. Our experiments show that GEN-Z consistently improves classification performance over zero-shot baselines and performs on par with strong few-shot baselines. Further, we show that this approach allows personalizing predictions by incorporating contextual information from label descriptions.

## LIMITATIONS

The generalizability of our paper's findings to languages other than English may be limited since all the datasets used in our study are exclusively in English. We make simplifying assumptions about prior probabilities of labels and independence of labels and contextual factors, which may not always hold in practice. We acknowledge that certain demographic attributes in our work may not fully represent the entire population. For example, due to data availability, we only conducted experiments with binary gender (male/female) for user information, despite the existence of diverse genders. Additionally, the definition and categorization of social attributes in the datasets used in our experiments might predominantly reflect Western-centric perspectives, as the majority of the work involved in designing and creating such datasets aligns with Western-centric viewpoints. Lastly, while our experiments encompass a diverse range of text classification tasks, we have not evaluated its performance on other kinds of tasks like sentence pair classification, question answering, etc. We leave these evaluations for future wrok.

## ETHICS STATEMENT

Personalization presents complex ethical considerations, with both benefits and potential risks. On the one hand, models tailored to specific settings or groups can yield positive outcomes. However, these personalized models may inadvertently reinforce biases or result in discriminatory behavior if their performance is uneven across different groups. Moreover, privacy concerns arise as end users may be reluctant to have certain attributes or personal information, such as their sexual orientation or religion, considered by the model. We believe our approach of using label description can mitigate such ethical concerns, particularly in comparison to embedding-based personalization methods. By employing interpretable user information through label descriptions, our method fosters transparency and controllability throughout the entire personalization process. This mitigates potential issues related to privacy and allows users to have insight into how their information is used. Nevertheless, it is important to acknowledge potential cases of misuse, where individuals intentionally modify their user attributes to game the model and achieve desired labels. Such scenarios highlight the need for future research on mitigating abuse and maintaining the integrity of the personalization framework.

## ACKNOWLEDGEMENTS

Part of this work was done when S.K. was a PhD Student at Carnegie Mellon University and was supported by a Google PhD Fellowship. This research is also supported in part by the Office of the Director of National Intelligence (ODNI), Intelligence Advanced Research Projects Activity (IARPA), via the HIATUS Program contract #2022-22072200004. This material is also funded by the DARPA Grant under Contract No. HR001120C0124. We also gratefully acknowledge support from NSF CAREER Grant No. IIS2142739, NSF Grants No. IIS2125201, IIS2203097, and the Alfred P. Sloan Foundation Fellowship. The views and conclusions contained herein are those of the authors and should not be interpreted as necessarily representing the official policies, either expressed or implied, of ODNI, IARPA, or the U.S. Government. The U.S. Government is authorized to reproduce and distribute reprints for governmental purposes notwithstanding any copyright annotation therein.

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

## A CONNECTION TO SURFACE FORM COMPETITION

Discriminative classification from language models have been shown to suffer from "surface form competition" where multiple surface forms of the label $y_i$ may compete for probability mass. To address this issue, Holtzman et al. (2021); Zhao et al. (2021) proposed calibrating the probability by using point-wise mutual information (PMI) between the text and the label as the scoring function, which is given as, $\hat{y} = \arg\max_{y_i} \frac{p(\mathbf{x}, y_i)}{p(\mathbf{x})p(y_i)}$. While these works have simplified PMI as $p(y_i|\mathbf{x})/p(y_i)$, an alternative way to simplify it is $p(\mathbf{x}|y_i)/p(\mathbf{x})$ which is the same as the generative classification setup as described above.

## B ADDITIONAL EXPERIMENTAL DETAILS

**Discriminative Baselines Variations** For each discriminative zero-shot baseline, we consider three variations (see Table 6): (1) DISC-NONE does not condition the input text on contextual variables, (2) DISC-CONTEXT conditions the input text on the contextual variables using a simple format, (3) DISC-INSTRUCT conditions the input text on the contextual variables as well as the labels that the model is expected to predict. In the main paper, we present results with DISC-NONE, which performs best out of these variations.

Table 4 summarizes all the datasets we use. Table 5 summarizes the hand written templates we start with (and later paraphrase to construct label descriptions).

| Dataset | Task (Domain) | #Classes |
|---|---|---|
| SST2 (Socher et al., 2013) | Sentiment Classification (movie) | 2 |
| SST5 (Socher et al., 2013) | Sentiment Classification (movie) | 5 |
| Yelp2 (Zhang et al., 2015a) | Sentiment Classification (Yelp) | 5 |
| Yelp5 (Zhang et al., 2015a) | Sentiment Classification (Yelp) | 5 |
| Poem Sentiment (PS) (Sheng & Uthus, 2020) | Sentiment Classification (Poetry) | 4 |
| Financial Phrasebank (FP) (Malo et al., 2014) | Sentiment Classification (Economic News) | 3 |
| MR (Pang & Lee, 2005) | Sentiment Classification (Rotten Tomatoes) | 2 |
| CR (Hu & Liu, 2004) | Sentiment Classification (Customer Reviews) | 2 |
| Tweet3 (Rosenthal et al., 2017) | Sentiment Classification (Twitter) | 3 |
| AGNews (Zhang et al., 2015b) | Topic Classification (News) | 4 |
| DBpedia | Topic Classification (Wikipedia) | 14 |
| Hate_speech18 (de Gibert et al., 2018) | Hate Speech (Stormfront) | 2 |
| Ethos (4 subsets) (Mollas et al., 2022) | Hate Speech by Subject (various social media) | 2 |
| Emotion (Saravia et al., 2018) | Emotion Recognition (Twitter) | 6 |
| Potato Prolific (Pei & Jurgens, 2023) | Politeness Classification (Email) | 2 |
| Talk Up (Njoo et al., 2023) | Empowerment prediction (Reddit) | 2 |

Table 4: Datasets used for the experiments

| Task | Label Description |
|---|---|
| Sentiment | "This [DOMAIN] leans [POLARITY]: "; DOMAIN∈{text, movie review, RottenTomatoes review, tweet, customer review, Yelp review, poem verse, financial news excerpt}, POLARITY∈{very positive, positive, neutral, negative, very negative}. We use "positive" and "negative" as POLARITY for binary sentiment classification. |
| Hate speech | "This [DOMAIN] uses [LABEL] language: "; DOMAIN∈{text, Stormfront post}, LABEL∈{hateful, innocuous}. |
| Ethos | "This [DOMAIN] contains hate-speech about [SUBJECT]: "; DOMAIN∈{text, social media post}, SUBJECT∈{something, national origin, religion, race, sexual orientation}. This is a binary classification task where "something" serves as the negative class for every other subject. |
| Topic | "The topic of this [DOMAIN] revolves around [TOPIC]: "; DOMAIN∈{text, news excerpt}, TOPIC∈{world, sports, business, technology} for AGNews, TOPIC∈{company, educational institution, artist, athlete, office holder, means of transportation, building, natural place, village, animal, plant, album, film, written work} |
| Politeness | "According to a [AGE] years old person with a [EDUCATION], this email snippet is impolite:"; AGE∈ set of integers, EDUCATION∈{High school degree, college degree, }. Both are provided in the test example. |
| Emotion | "This [DOMAIN] emotes [EMOTION]: "; DOMAIN={text, tweet}, EMOTION={sadness, love, anger, joy, fear, surprise} |
| Empowerment | "This Reddit comment written by a [AUTHOR] empowers and uplifts the addressed [ADDRESSEE]: "; AUTHOR={man, woman}, ADDRESSEE={man, woman} |

Table 5: Label description starter templates we hand write. DOMAIN="text" represents missing domain information (which is one of our ablation). We generate their variations by asking ChatGPT: "Write 11 paraphrases of this sentence as a Python list." The full list is provided in the supplementary material.

## C  ADDITIONAL RESULTS

We provide averaged macro-F1 for each of the models and each zero-shot method we consider in Figures 3,4,5,6,7, 8, 9, 10

| Name | Format |
|---|---|
| DISC-NONE | "[INPUT] [LABEL DESCRIPTION]" |
| DISC-CONTEXT | "This is a [DOMAIN]. [INPUT] [LABEL DESCRIPTION]" |
| DISC-INSTRUCT | "Is this [DOMAIN] [LIST OF LABEL NAMES]? [INPUT] [LABEL DESCRIPTION]" |

Table 6: Different variations of DISC models. The results reported in the main paper are from DISC-NONE as it performed the best overall. In all three settings the label descriptions always contain the contextual information. The results for all three approaches can be found in Figures 3,4,5,6,7, and 8

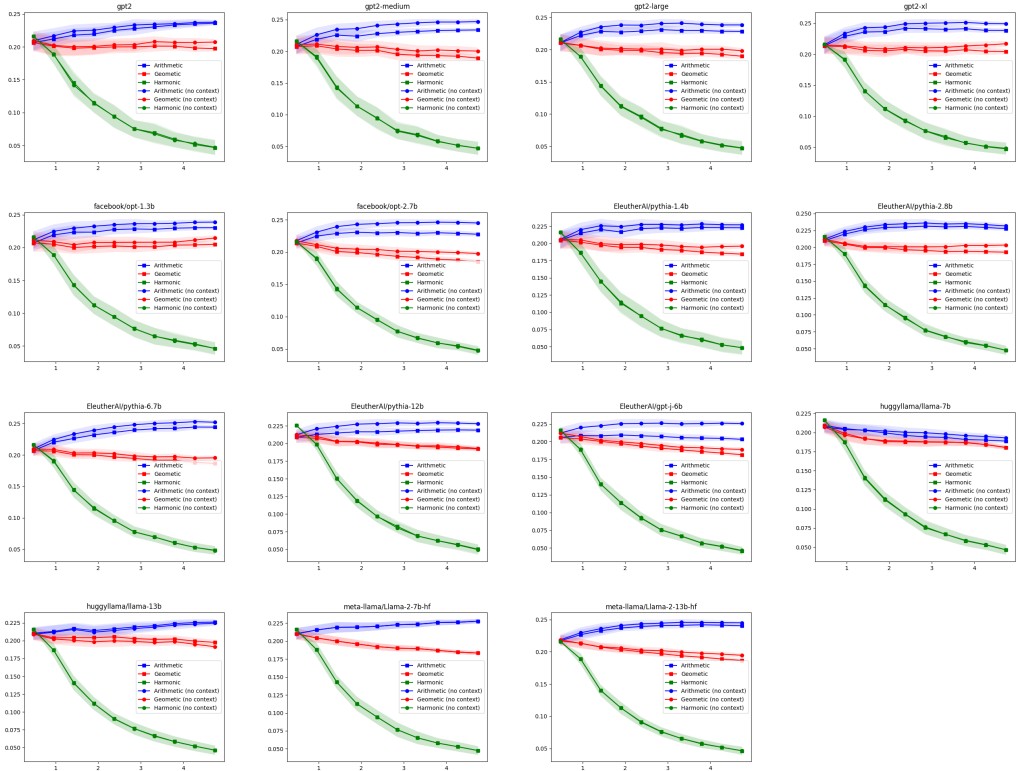

Figure 3: Full results for domain aware classification for our zero-shot DISC-NONE setup without calibration. The x-axis shows number of label descriptions per label and the y-axis indicates the average accuracy across all the tasks except (Politeness and Empowerment). We conduct a thorough analysis of this setup: removing domain information from the descriptions, different aggregation strategies as well as evaluating on different model sizes and families.

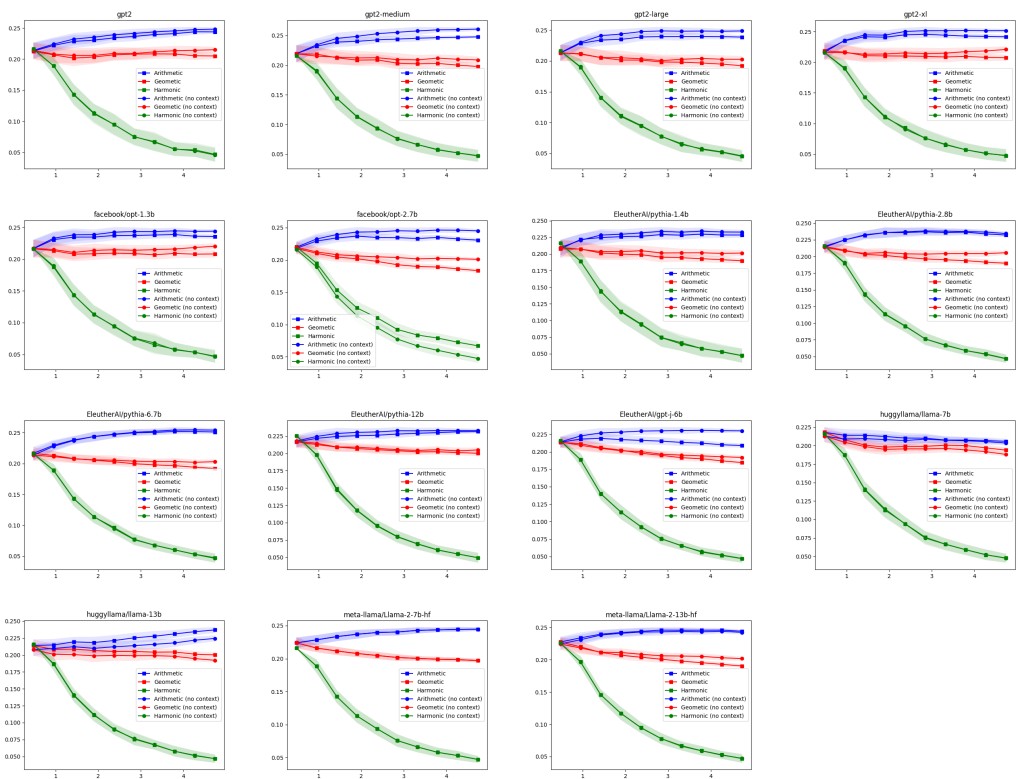

Figure 4: Full results for domain aware classification for our zero-shot DISC-CONTEXT setup without calibration. The x-axis shows number of label descriptions per label and the y-axis indicates the average accuracy across all the tasks except (Politeness and Empowerment). We conduct a thorough analysis of this setup as discussed in (section 4): removing domain information from the descriptions, different aggregation strategies as well as evaluating on different model sizes and families.

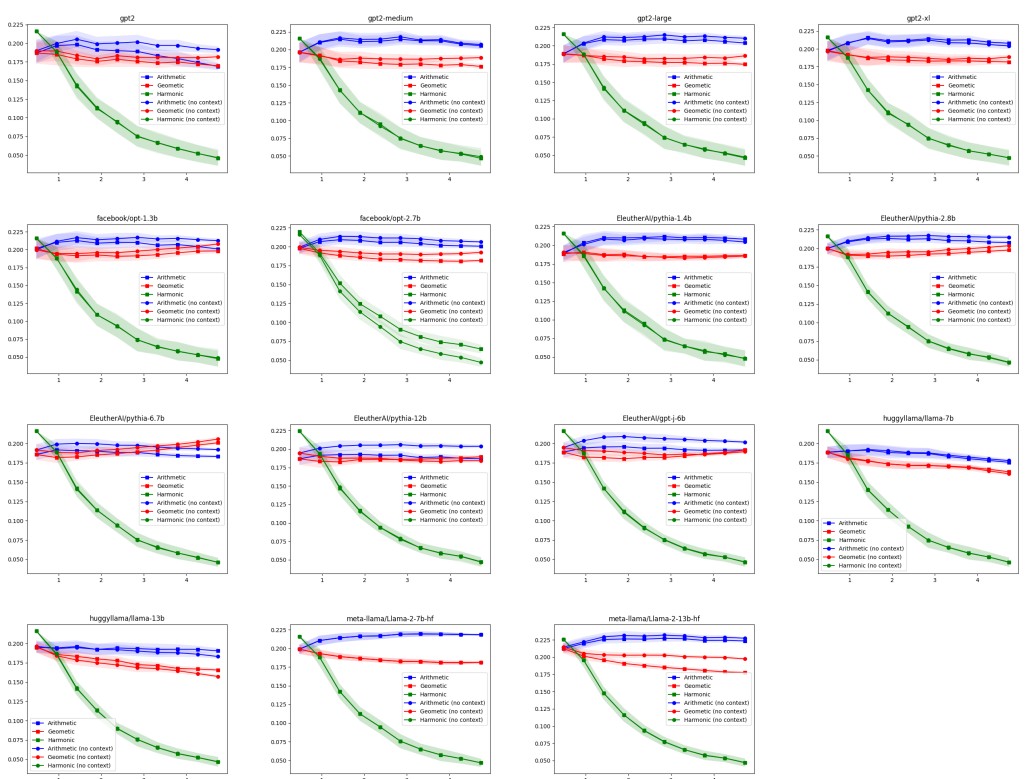

Figure 5: Full results for domain aware classification for zero-shot DISC-INSTRUCT setup without calibration. The x-axis shows number of label descriptions per label and the y-axis indicates the average accuracy across all the tasks except (Politeness and Empowerment). We conduct a thorough analysis of this setup as discussed in (section 4): removing domain information from the descriptions, different aggregation strategies as well as evaluating on different model sizes and families.

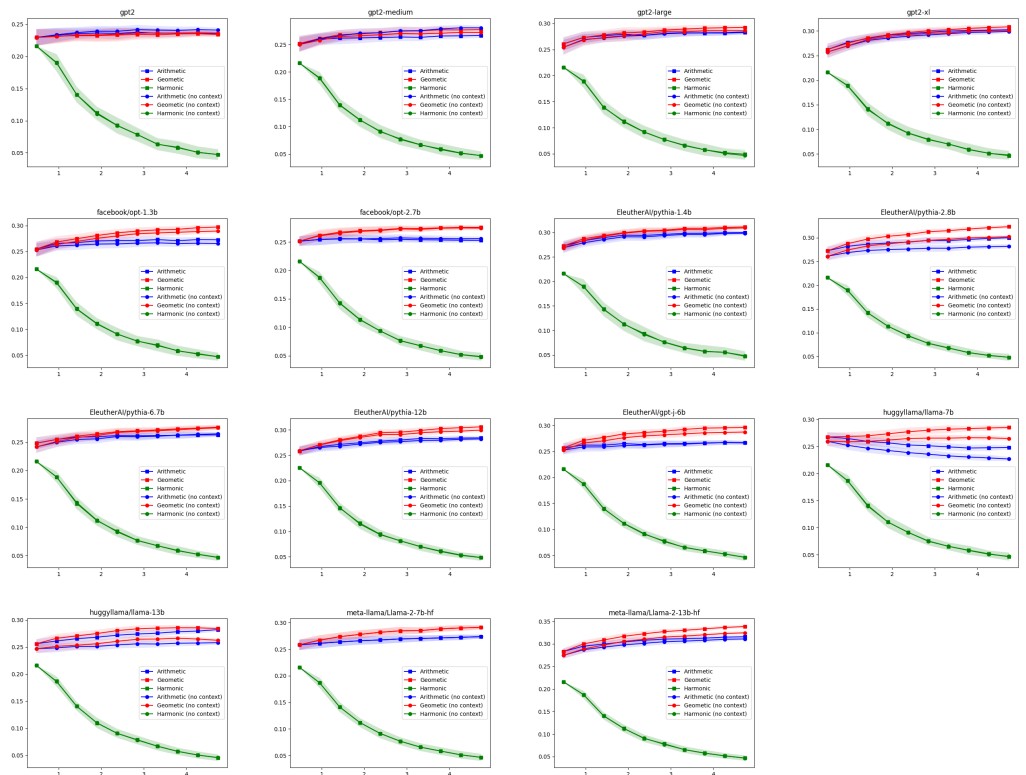

Figure 6: Full results for domain aware classification for our zero-shot DISC-NONE setup with PMI based calibration. The x-axis shows number of label descriptions per label and the y-axis indicates the average accuracy across all the tasks except (Politeness and Empowerment). We conduct a thorough analysis of this setup as discussed in (section 4): removing domain information from the descriptions, different aggregation strategies as well as evaluating on different model sizes and families.

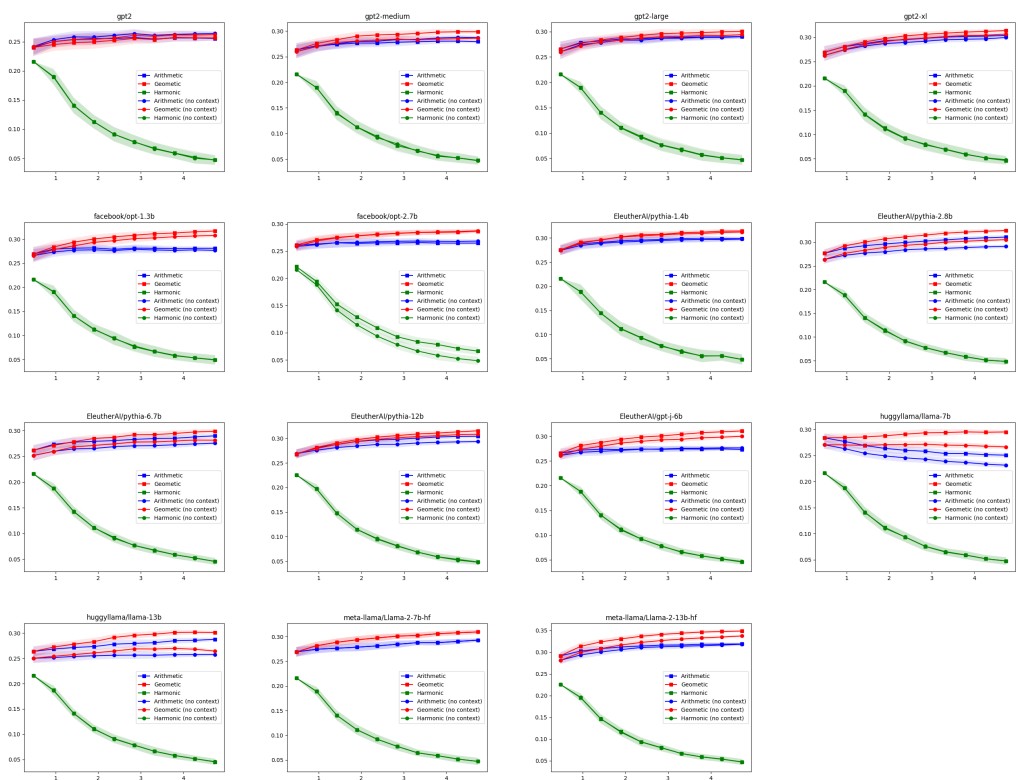

Figure 7: Full results for domain aware classification for our zero-shot DISC-CONTEXT setup with PMI based calibration. The x-axis shows number of label descriptions per label and the y-axis indicates the average accuracy across all the tasks except (Politeness and Empowerment). We conduct a thorough analysis of this setup as discussed in (section 4): removing domain information from the descriptions, different aggregation strategies as well as evaluating on different model sizes and families.

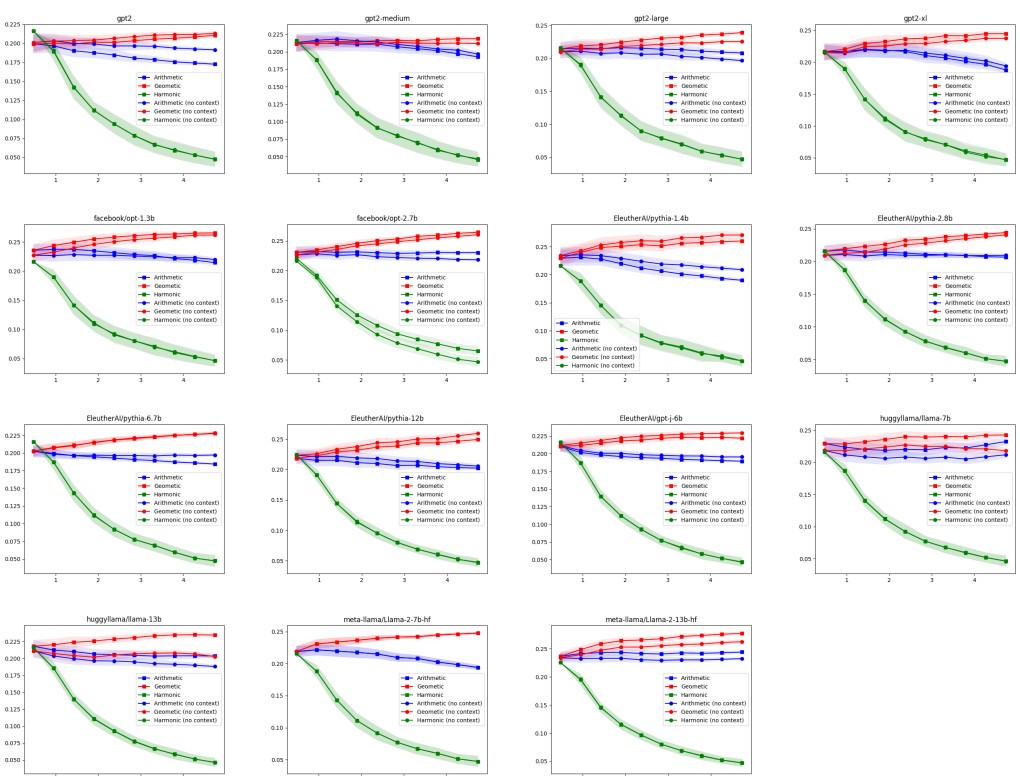

Figure 8: Full results for domain aware classification for our zero-shot DISC-INSTRUCT setup with PMI based calibration. The x-axis shows number of label descriptions per label and the y-axis indicates the average accuracy across all the tasks except (Politeness and Empowerment). We conduct a thorough analysis of this setup as discussed in (section 4): removing domain information from the descriptions, different aggregation strategies as well as evaluating on different model sizes and families.

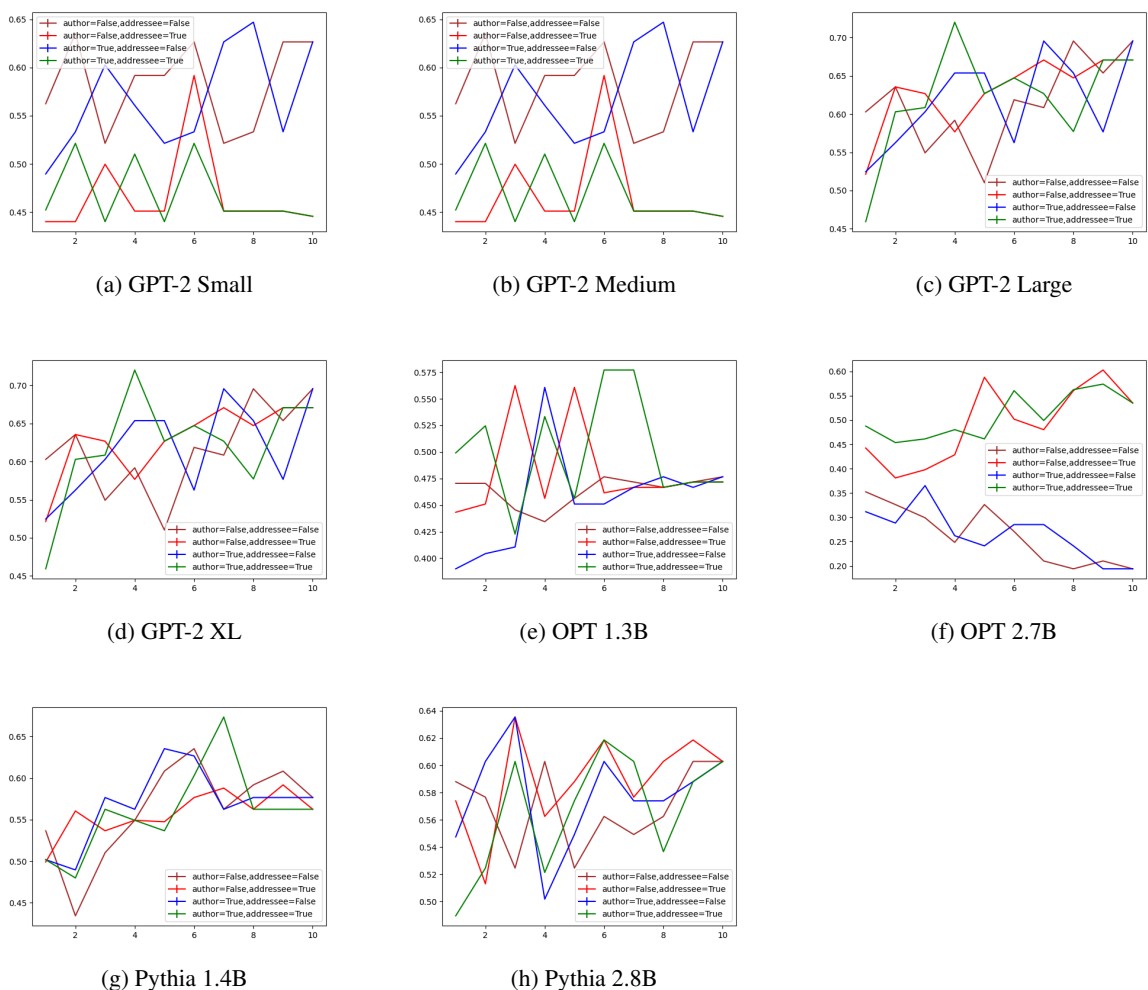

Figure 9: Full results for author and addressee-personalized empowerment prediction with our proposed setup. The x-axis shows number of label descriptions per label and the y-axis indicates the average F1-score. The four plots indicate 4 settings described in Table 3.

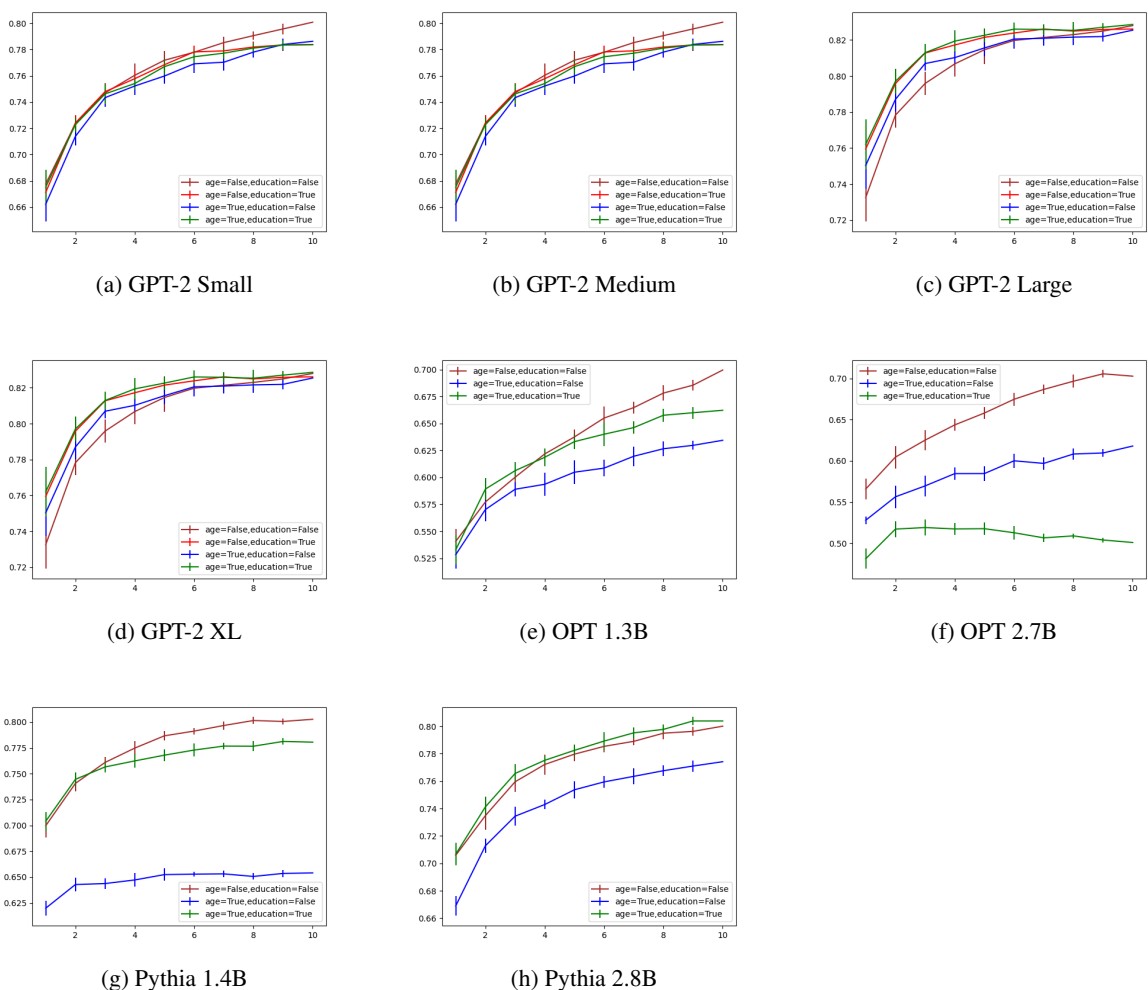

Figure 10: Full results for reader-personalized politeness prediction with our proposed setup. The x-axis shows number of label descriptions per label and the y-axis indicates the average accuracy. The four plots indicate 4 settings described in Table 3.

