# OpenReview forum: "Gen-Z: Generative Zero-Shot Text Classification with Contextualized Label Descriptions"
_ICLR.cc/2024/Conference — ICLR 2024 poster_

### Official Review · Reviewer_N7Hu · 2023-10-27

**Soundness:** 3 good
**Presentation:** 3 good
**Contribution:** 2 fair
**Rating:** 6
**Confidence:** 3

**Summary:**

This paper proposes a generative prompting framework for zero-shot text classification. Specifically, the proposed model takes on the text classification task by measuring the LM likelihood of the input text, conditioned on natural language descriptions of labels.GEN-Z leverages LM likelihood of generating the input text based on various label descriptions that reflect context, enabling more robust
predictions than discriminative approaches. Experiments show that GEN-Z consistently improves classification performance
over zero-shot baselines and performs on par with strong few-shot baselines

**Strengths:**

1) This generative in-context learning is natural due to the LLM being based on a generative model. This paper propose two novel module based on baseline models, including MULTIVARIATE GENERATIVE CLASSIFICATION and CONTEXTUALIZED LABEL DESCRIPTIONS.

2) Compared with other in-context learning methods, this method can improve robustness effectively.

3) The paper is well written, and the figure is helpful for understanding.

4) The analysis of the Baysian equation is novel, simple, and clear.

**Weaknesses:**

This paper may take incremental work, which adds two new modules based on the generative classifier in [1].
In my opinion, the work doesn't show any new insight, maybe not enough for ICLR but more suitable for other conference , such as ACL.

[1] Min, Sewon, et al. "Noisy channel language model prompting for few-shot text classification." arXiv preprint arXiv:2108.04106 (2021).

**Questions:**

Do you add the label description to other baseline models, which are generated from ChatGPT? If not, the comparison may not be fair because you added more external knowledge.

Can the generative classifier be used to other suan as multimodel, or image data?(CLIP)

---

> ### Author Response · Authors · 2023-11-22
> **Response to Reviewer N7Hu**
>
> We thank the reviewer for their feedback and highlighting the simplicity and the novelty of our method and the clear writing of the paper. Please find our response to your questions and concerns below:
>
> > the work doesn't show any new insight, suitable for NLP venues like ACL.
>
> We disagree that this work is only suitable for NLP venues. A large majority of work presented at ICLR includes work in improving/understanding in-context learning and our work falls under that umbrella. The main contribution of our work is to show that zero-shot classification is a strong method competitive or even better than in-context learning. Zero-shot classification is usually considered a weak baseline and has been shown to perform substantially worse than in-context methods. We showed that (1) pretrained LMs can indeed perform strongly zero-shot and it should not be discarded as a weak baseline but in fact be the only method to consider without worrying about in-context examples which can be hard to find for some tasks such as those involving personalized attributes, and, (2) shine a light on why in-context learning might be performing as well as it does. In section 2.1, we build on previously suggested theories that in-context learning learns concepts from examples and build on that idea to present our method.
>
> > Do you add the label description to other baseline models, which are generated from ChatGPT? If not, the comparison may not be fair because you added more external knowledge.
>
> The label descriptions we consider in our approach are simple paraphrases of a templated label description we hand wrote. ChatGPT is used to generate these paraphrases due to its ease of use. The paraphrases do not add any external knowledge to the inference process. This process is done without relying on any development data. The zero-shot baselines we consider are meant to show the importance of considering multiple descriptions and are in way ablations of our approach, where removing multiple label descriptions hurt the performance (table 1; columns GS, DS, and DPS). Further, combining few-shot methods with multiple label descriptions did not yield improvements over existing few-shot setups in our initial explorations. Combining few-shot methods with multiple label descriptions is non-trivial and we leave that exploration to future work.
>
> > Can generative classification be used in multimodal settings
>
> Some recent work on arXiv [1] has shown that, indeed, diffusion models for text to image generation can be used to classify images and have many interesting properties such as better robustness and domain generalizability.
> [1] https://arxiv.org/abs/2309.16779

---

### Official Review · Reviewer_BREF · 2023-10-30

**Soundness:** 3 good
**Presentation:** 3 good
**Contribution:** 3 good
**Rating:** 6
**Confidence:** 4

**Summary:**

This work proposes GEN-Z, a generative prompting framework for zero-shot text classification. Specifically, GEN-Z measures the LM likelihood of input text, conditioned on natural language descriptions of labels, which can be extended to multivariate forms. This work conduct experiments on multiple classification tasks. Experimental results show that the proposed framework can achieve better performance compared with the baseline approaches.

**Strengths:**

1. The proposed framework is interesting and seems novel. Utilizing label description in the bayesian form can help to improve the overall classification performance as shown in the experiments in the paper.

2. This work conducts extensive experiments on a variety of datasets, including domain-aware classification and personalized classification. This is very encouraged in NLP research.

**Weaknesses:**

1. Though this work argues that utilizing label descriptions in a bayesian form benefits the performance, it relies heavily on the quality of the label description, e.g., how clear the context can present the task, domain the label. However, in text classification, sometimes the label is not natural language, but a less literally meaningful symbol. For such labels, it is not likely to encourage the model to generate text $x$ given $y$. Then it requires human beings to provide very good summaries or descriptions for such labels, which is sometimes challenging. Such a problem limits the application of the proposed GEN-Z framework. Instead, conventional in-context learning can provide examples and have the model to learn from simpler label descriptions and demonstrations, which is more flexible.

2. Since label descriptions directly decides how the model generates $x$, the quality of $z_i$ plays an important role on the performance. It is encouraged to explore how the performance is influenced if descriptions of different quality are given. For example, if the label can already express the meaning of the label, then descriptions may not be required. When the label meaning is more abstract, detailed descriptions may be better than abstract descriptions. Such influence is encouraged to be explored.

**Questions:**

1. For LMs, the context can influence the language model probability significantly. Combining multiple label descriptions may result in changed language model probability. So how is multiple label descriptions utilized in this work?

---

> ### Author Response · Authors · 2023-11-22
> **Response to Reviewer BREF**
>
> We thank the reviewer for their positive feedback and highlighting the novelty of our method and the extensiveness of our experimental setup. Please find our response to your questions and concerns below:
>
> > Difficult to describe context for some tasks.
>
> Thank you for pointing this out. It is possible that some tasks may require more descriptive labels than the ones we considered. But, for any text classification task where annotations are involved in creating the training/development/test sets, the prevalent approach is to provide detailed instructions to the annotators including describing what each label is supposed to capture. That is to say,  human labor is already involved in describing labels in most text classification tasks. Such descriptions can easily be reused to work with our setup if available publicly. It is an interesting avenue for future work to study this behavior. We will include this in the limitations section in the updated draft.
>
> Furthermore, the utility of our method lies in cases where few-shot examples may be not be straightforward to obtain. There are several tasks where it is not possible to access any labeled data to create few-shot examples such as those discussed in our personalization classification experiments. For example, when we’d like to classify text on dialects, but also social variations spoken by intersectional identities (e.g. an older, non-binary, educated speaker of New-Zealand English), finding annotators who can accurately annotate such data is very challenging. Despite requiring potentially long descriptions, it is possible to use Gen-Z in these settings.
>
> >For LMs, the context can influence the language model probability significantly. Combining multiple label descriptions may result in changed language model probability. So how is multiple label descriptions utilized in this work?
>
> Multiple label descriptions are simple paraphrases of each other (we provide all the descriptions we used here:  https://pastebin.com/UwH8zRuV). It is indeed true that conditioning on different paraphrases can result in different text probabilities. We propose to simply add the conditional probabilities corresponding to each label and then compare the aggregated probabilities to predict the label which results in improvement in mean performance and reduction in the performance variance. We also ablate with other aggregation methods such as multiplying them or computing their harmonic mean (reported in the Appendix), but overall found simple summation to perform consistently well.

---

### Official Review · Reviewer_uHMN · 2023-10-30

**Soundness:** 3 good
**Presentation:** 3 good
**Contribution:** 2 fair
**Rating:** 6
**Confidence:** 4

**Summary:**

This study suggests employing multivariate generative classification over discriminative approaches for zero-shot text classification tasks utilizing in-context learning.
Building on Min et al. (2022), the paper demonstrates that through multiple iterations with varied contexts elucidating label information, generative classification coupled with in-context learning emerges as a more rational and stable solution.
Furthermore, users can enhance the context with additional information (e.g., the gender of a writer) to more effectively clarify the label space.
It is evidenced that this supplementary context can significantly improve the performance of in-context learning in zero-shot classification tasks.

**Strengths:**

- The authors successfully derived the final form of the probability for generative text classification (i.e., $\sum_{z_i} p(\mathbf{x}|z_i)$), based on a series of reasonable assumptions.
- The paper outlines a well-considered experimental setup that demonstrates the efficacy of the proposed method in specific applications like domain-aware classification and personalized classification.

**Weaknesses:**

- The method proposed is conceptually similar to Min et al.'s noisy channel method, with the primary distinction being the use of multiple "label descriptions". This similarity raises questions regarding the novelty of the proposed approach.
- The assumptions outlined in Section 2.3 may be overly simplistic, potentially detracting from the final performance of the proposed method. The paper would gain interest if the authors could suggest reasonable approximations for the omitted probabilities, namely i.e., $p(z|y_i, u,v,\dots)$ and $p(y_i|u,v,\dots)$.

**Questions:**

- What potential impacts could arise from the combination of the proposed method with few-shot learning scenarios? This consideration is crucial, especially given that while generative zero-shot classification yields reasonable results, it does not yet match the performance of few-shot-based methods, like ICL (DC) as depicted in Table 2.

---

> ### Author Response · Authors · 2023-11-22
> **Response to Reviewer uHMN**
>
> We thank the reviewer for their feedback and highlighting the clarity of our mathematical framework, thorough experimental setup, efficacy of our method and overall soundness. Please find our response to your questions and concerns below:
>
> > conceptually similar to Min et al.'s noisy channel method
>
> This paper serves as a source of inspiration for our work, but we make two simple but significant changes to their method, which lead to substantial improvements over Min et al’s performance. By incorporating multiple label descriptions and adding contextual information to this task, Gen-Z outperforms Min et al’s performance in both zero- and few-shot versions. These changes, albeit simple, are non-trivial and have theoretical grounding. Our method is easily reproducible and usable by others, especially since it is zero-shot. In addition to prompt variations, prior work has shown that selection of in-context examples has a significant impact on task performance. Not requiring any in-context examples at all and is a significant contribution.
>
> > assumptions in section 2.3
>
> Thank you for pointing this out. The assumptions made in this section serve to simplify the inference objective and have sound motivations. We have clarified them in the paper further, we repeat them here.
>
> In the graphical model that we design as shown in Figure 1 (right), each conditional represents the true probability. Give the label $y$ and attributes $u, v,  …$ if there are $N$ ways in which we can describe them in textual form, $p(z|....)$ should really be equal for each possible paraphrase, we are not defining the grammatical plausibility of the paraphrase, just the existence probability, hence using the LM’s probably for $p(z|...)$ is not appropriate.
>
> For $p(y | u, v, …),$, first, we assume that the probability of the label is independent of contextual factors. For most factors we consider in this work, this assumption is largely true. The true prior probability of any label is unlikely to depend on the domain of the text, or the personal attributes of the user reading, writing, or being described in the text (we agree that this might not hold in datasets on which classifiers are typically trained). Second, we assume that each label is equally likely, this is a simplifying assumption and may not hold in practice but it is not trivial to estimate this using an LM. Our initial exploration in this direction showed decline in classification performance.
>
> > Combining Gen-Z with few-shot prompting
>
> This is a great suggestion and avenue for future work. Zero-shot classification is usually considered a weak baseline and has been shown to perform substantially worse than in-context methods. Our goal is in this paper to two fold (1) make a case that pretrained LMs can indeed perform strongly zero-shot and it should not be discarded as weak baseline, and, (2) shine a light on why in-context learning might be performing as well as it does. We do not claim to build a new state of the art prompting method or suggesting that in-context learning is not necessary. In our initial exploration, combining our approach with few-shot examples trivially did not show improvements over either methods, most likely due to the label description getting ignored due to long context between the test example and the description. But alternative methods of incorporating in-context examples such as ensembling might be interesting to explore in future work.
>
> Further, we would like to point out that the ICL method DC is not strictly comparable to our approach as in addition to few-shot examples, it requires access to a several unlabeled examples in the same domain as the test set to be able to compute calibrated LM probabilities which gives them a boost in performance. We do not assume access to such additional dataset in our work.

---

### Official Review · Reviewer_vWvw · 2023-11-01

**Soundness:** 3 good
**Presentation:** 3 good
**Contribution:** 3 good
**Rating:** 6
**Confidence:** 3

**Summary:**

This paper introduces a generative prompting framework for zero-shot text classification. It proposes to make text classification based on the LMs' likelihood of input text conditioned on the description of each label, where the label description comes from human annotation and ChatGPT. Experiments show this proposed model achieves solid zero-shot text classification performance over baselines.

**Strengths:**

- This paper is well-written and easy to follow.
- The empirical analysis is comprehensive and solid regarding datasets and backbone models.
- The proposed method is simple yet effective, providing a solid and lightweight framework to solve zero-shot text classification problems.

**Weaknesses:**

The proposed method, presentation, and empirical analysis are well self-explained. I don't have much concern about it.

My main concern is whether the problem of "zero-shot text classification with some human/ChatGPT labeled data" is meaningful. If one would like to do some annotation, either with human annotators or ChatGPT, why not directly annotate some labeled samples for few-shot text classification? Thus, in my opinion, to show that this problem setup and the proposed framework are meaningful, the authors need to somehow show that this zero-shot framework is more effective than a few-shot text classification framework given the same amount of annotation effort.

**Questions:**

Do you think the proposed framework can be used to amplify few-shot classification systems? I think it may make better sense in assisting a few-shot classification system since adding a few more training samples may not be as effective as providing these label descriptions given the same annotation efforts.

---

> ### Author Response · Authors · 2023-11-22
> **Response to Reviewer vWvw**
>
> We thank the reviewer for their feedback and highlighting that our paper is well written, easy to follow and presents a simple, lightweight, solid and effective approach. Please find our response to your questions and concerns below:
>
> > If one would like to do some annotation, either with human annotators or ChatGPT, why not directly annotate some labeled samples for few-shot text classification?
>
> There are several tasks where it is not possible to access any labeled data to create few-shot examples such as those discussed in our personalization classification experiments. For example, when we’d like to classify text on dialects, but also social variations spoken by intersectional identities (e.g. an older, non-binary, educated speaker of New-Zealand English), finding annotators who can accurately annotate such data is very challenging. Gen-Z is easy to use in such settings.
>
> Additionally, compared to few-shot setups where the knowledge of the task is needed, our proposed setup is much easier to implement. First, the human effort required in this work is minimal. We hand write the description of each potential label for each task which does not require any deep knowledge of the task and largely follows a template:
>
> `This is a {context information} with {label_0}`
>
> We have provided the descriptions we handwrote according to these templates in Table 5 in the Appendix. This is a comparable setting to any zero-shot classification setup where the model is prompted in some way including providing the name of the task.
> Second, we only use ChatGPT to generate paraphrases of the label descriptions we handwrote. The paraphrases we used to generate our results can be found in this pastebin link: https://pastebin.com/UwH8zRuV. ChatGPT is not meant to inject any knowledge into this model but rather providing very simplistic paraphrases of the label descriptions. Importantly, ChatGPT is used for convenience but is not crucial to the success of our proposed method. Any simple paraphrasing model can be employed, even template based ones. As an example, for the GPT-J (6B) model, we recomputed the results for our proposed approach using paraphrases generated using beam search by a much smaller model trained to generate only paraphrases (tuner007/pegasus_paraphrase; 568M) and were able to obtain mean F1 scores to the reported results in the paper sometimes even outperforming ChatGPT paraphrases (see results below). We will update the full set of experiments with all the model families in the final draft given more time.
>
> Paraphraser | SST2 | SST5 | Tweet3 | AGNews | HateSpeech18 | Emotions
> -------------- |--------|--------|---------|------------|------------------|------------
> ChatGPT      | 91.7   | 40.9   | 41.1     | 77.0         |  62.6                 | 32.7
> Pegasus       |  91.4  |  40.3  |  40.8    | 76.8         |  64.0                 | 33.9
>
>
>
>
> > Do you think the proposed framework can be used to amplify few-shot classification systems?
>
> In our initial exploration, we did not observe significant improvements combining few-shot learning with label descriptions, especially with 8-16 shot setups where we observed that the label descriptions are essentially ignored due to the long distance between the description and the test example. Min et al 2022 shows that reformulating k-shot learning as an ensemble for k 1-shot learning setups can lead to improvements in certain cases but their results were inconclusive. We agree that it would be a valuable next step to combine our proposed approach with in-context learning but it is non-trivial to make it work and we leave that exploration to future work.

---

### Meta-Review · Area_Chair_95Mu · 2023-12-10

**Metareview:**

In this paper, the authors introduce GEN-Z, a robust generative zero-shot text classification framework that incorporates contextual information beyond the input text itself. Specifically, GEN-Z leverages LM likelihood of generating the input text based on various label descriptions that reflect context, enabling more robust predictions than discriminative approaches. The experimental results show the effectiveness of the proposed framework.

The paper is well-written and easy to follow. The experiments are well set-up and the results are good. However, the novelty of the paper is somewhat limited. During the rebuttal, the reviewers had several questions about the technical details and potential impact of the work, and the authors addressed them well in the response.

**Justification For Why Not Higher Score:**

The novelty of the paper is somewhat limited

**Justification For Why Not Lower Score:**

The paper is well-written and the motivation is good

---

### Decision · Program_Chairs · 2024-01-16

Accept (poster)